# Measuring Procedural Reuse Judgments with Response Curves

## Abstract

Procedural knowledge such as recipes, protocols, and workflows is often adapted rather than executed directly. Understanding such reuse judgments requires analyzing how alternative procedures are comparatively evaluated under different procedural and contextual conditions.

We introduce a measurement framework for studying *comparative reuse judgments*, defined as decisions about which of two candidate procedures is preferred for reuse with respect to a reference procedure under a given context. We construct controlled differences between candidate procedures along predefined structural axes and aggregate pairwise decisions into *response curves*, which characterize how alignment probabilities vary across levels of procedural contrast.

We instantiate this framework in the domain of cooking recipes and use a large language model as a controlled decision-maker to generate comparative judgments at scale. The resulting response curves exhibit structure beyond randomized controls, reveal both aggregate and reference-specific variability, and undergo measurable shifts under contextual interventions. The framework further enables systematic comparison of response-curve structures across different decision-makers.

These findings suggest that comparative reuse judgments are more naturally characterized as response functions over procedural contrasts rather than as single similarity scores or global preference measures.

More broadly, this work proposes a measurement-oriented framework for studying comparative reuse judgments, enabling systematic analysis and comparison of response-curve structures under controlled procedural and contextual variation.

## 1 Introduction

Procedural knowledge is commonly shared through artifacts such as cooking recipes, experimental protocols, and operational workflows.

A large body of work has focused on representing and interpreting procedural knowledge in structured, machine-readable forms. This includes the interpretation of instructional texts such as recipes (Kiddon et al., 2015), machine reading of wet-lab protocols (Kulkarni et al., 2018), formal semantic representations of biomedical protocols (Soldatova et al., 2008; 2014), and workflow modeling and process mining (van der Aalst & van Hee, 2002; Van Der Aalst, 2011). More recent work has explored the automatic construction of domain-specific representations of procedural knowledge (Shi et al., 2024).

These approaches primarily aim to support representation, interpretation, prediction, or execution. However, procedural reuse often involves a different type of decision problem: determining which alternative procedure is more applicable as a source for adaptation relative to a particular procedure and context.

Execution-based evaluation captures one aspect of procedural reuse, focusing on whether a procedure can be successfully applied. However, many real-world reuse decisions arise before execution, when alternative

procedures are assessed for their potential applicability under adaptation. Evaluations based solely on execution outcomes therefore provide limited insight into how procedural alternatives are comparatively evaluated for reuse.

A key challenge is that such reuse judgments are inherently reference- and context-conditioned. The applicability of a candidate procedure depends not only on its own characteristics, but also on the procedure currently being adapted and the constraints under which adaptation is considered.

Our objective is to characterize how comparative reuse judgments vary under controlled procedural contrasts and contextual conditions. We therefore focus on making comparative reuse judgments observable and measurable through controlled comparisons, rather than treating reuse as a prediction, utility estimation, or execution-evaluation problem.

We define *reuse judgments* as decisions about which of two candidate procedures is preferred for reuse with respect to a reference procedure under a given context. This formulation operationalizes comparative applicability assessments relative to an adaptation target, allowing comparative reuse judgments to be studied through observable decisions.

A central aspect of our approach is the introduction of *structural axes*, each of which defines a particular dimension of procedural variation (e.g., step similarity, ingredient similarity, ingredient rarity, or recipe style). These axes serve as measurement coordinates that define controlled dimensions along which comparative reuse judgments can be observed and analyzed.

For each axis, we define *contrast* as the difference between the reference-relative comparison values of the two candidate procedures along that axis. Importantly, contrasts are defined relative to a reference procedure rather than as positions in a global procedural space. By operating within an existing procedural corpus, we can vary these contrasts in a controlled manner while holding other factors fixed, enabling systematic comparison across multiple axes within a shared procedural context.

Under this framework, individual decisions are treated as observations of comparative reuse judgments under controlled conditions. By aggregating decisions across different levels of contrast, we obtain response patterns that characterize how reuse judgments vary with contrast along each axis, without assuming a fixed or monotonic relationship. In practice, contrasts are grouped into bins, allowing response curves to be estimated as functions of contrast.

This perspective is related to classical models of comparative judgment (Thurstone, 1927) and to measurement paradigms in psychophysics, where response probabilities are analyzed as functions of controlled stimulus differences (Green & Swets, 1966; Wichmann & Hill, 2001).

We implement this framework using a large language model (LLM) as a decision-maker to generate comparative judgments at scale under controlled conditions, following prior work on using LLMs as evaluators (Zheng et al., 2023; Liu et al., 2023).

We instantiate the framework in the domain of cooking recipes. While recipes represent only one class of procedural knowledge, they provide a structured setting with naturally occurring goals, resources, constraints, and adaptation scenarios, making them suitable for constructing controlled procedural contrasts and studying context-conditioned reuse judgments.

In this setting, we construct candidate procedures that vary along structural axes such as step similarity, ingredient similarity, ingredient rarity, and style variation. Aggregating decisions within contrast bins yields response curves that characterize how comparative reuse judgments vary with contrast along each axis.

Our empirical analysis shows that response curves exhibit structure beyond randomized controls, differ across structural axes, show substantial heterogeneity across reference procedures, undergo measurable changes under contextual interventions, and can be compared across different decision-makers under matched conditions.

Our main contributions are as follows:

- A measurement framework that operationalizes reuse judgments through controlled pairwise comparisons over naturally occurring procedural data.

- An analysis showing that aggregated pairwise decisions induce response curves that characterize how comparative reuse judgments vary with contrast magnitude, enabling controlled comparison across structural axes within a shared procedural corpus.

- An empirical characterization showing that response patterns vary across reference procedures, contextual conditions, and decision-makers, while remaining measurable within a common response-curve framework.

By separating reuse judgments from execution-based evaluation, this work provides a measurement-oriented perspective on procedural comparison. The framework supports descriptive analysis of how comparative reuse judgments vary under controlled procedural and contextual variation.

## 2 A Measurement Framework for Reuse Evaluation

### 2.1 Problem Setup

Let $\mathcal{R}$ denote a set of procedural artifacts. Each artifact $r \in \mathcal{R}$ represents a procedure described by structured information consisting of (i) a sequence of procedural steps and (ii) associated procedural attributes.

Depending on the domain, these attributes may include required resources (e.g., ingredients or tools) and descriptions specifying the intended goal of the procedure.

Our objective is to characterize how comparative reuse judgments vary across procedures, contexts, and structural contrasts.

A comparative reuse judgment refers to a decision about which candidate procedure is preferred for reuse with respect to a reference procedure under a given context. Because such judgments are inherently reference- and context-conditioned, they are treated here as comparative observations rather than as indicators of globally optimal reuse judgments.

Accordingly, the framework is designed to measure how reuse judgments vary under controlled procedural variation. The resulting response patterns provide a descriptive characterization of comparative reuse judgments.

In this work, we instantiate the framework in the domain of cooking recipes (Section 3).

### 2.2 Measurement Construction

For a reference procedure $r_q \in \mathcal{R}$, a decision-maker is presented with two candidate procedures $\{r_1, r_2\} \subset \mathcal{R} \setminus \{r_q\}$, with $r_1 \neq r_2$, and selects the candidate that is *preferred for reuse* with respect to the reference $r_q$.

Each evaluation instance is represented as a trial

$$\tau = (r_q, r_1, r_2), \tag{1}$$

with a decision outcome

$$a(\tau) \in \{r_1, r_2\}, \tag{2}$$

where $a(\tau)$ denotes the candidate selected by the decision-maker. This formulation is invariant to the ordering or presentation of the candidates.

To relate these decisions to structural properties of procedures, we introduce a family of structural axes indexed by $k \in \mathcal{K}$. All axes are defined over the same underlying set of procedures $\mathcal{R}$, which serves as a shared procedural space for constructing controlled comparisons.

Each axis is defined by a feature mapping

$$\psi_k : \mathcal{R} \to \mathcal{Z}_k \tag{3}$$

that maps each procedure $r \in \mathcal{R}$ to an axis-specific representation $\psi_k(r) \in \mathcal{Z}_k$, together with a comparison function

$$\Delta_k : \mathcal{Z}_k \times \mathcal{Z}_k \to \mathbb{R}. \tag{4}$$

The framework does not assume that $\Delta_k$ satisfies metric properties such as symmetry or the triangle inequality. Depending on the axis, $\Delta_k$ may behave like a distance (e.g., semantic dissimilarity) or as a signed comparison value (e.g., differences in ingredient rarity).

For a trial $\tau = (r_q, r_1, r_2)$, we compute the reference-relative values

$$\delta_{k,1} = \Delta_k(\psi_k(r_q), \psi_k(r_1)), \qquad \delta_{k,2} = \Delta_k(\psi_k(r_q), \psi_k(r_2)), \tag{5}$$

and define the signed contrast

$$\tilde{x}_k(\tau) = \delta_{k,1} - \delta_{k,2}, \tag{6}$$

with magnitude

$$x_k(\tau) = |\tilde{x}_k(\tau)|. \tag{7}$$

The sign of $\tilde{x}_k(\tau)$ determines the candidate indicated by the structural comparison under axis $k$. Let

$$p_k(\tau) = \begin{cases} r_1 & \text{if } \tilde{x}_k(\tau) < 0, \\ r_2 & \text{if } \tilde{x}_k(\tau) > 0, \end{cases} \tag{8}$$

denote this candidate, where ties (i.e., $\tilde{x}_k(\tau) = 0$) are excluded from analysis.

Each axis therefore induces a direction of variation between candidates relative to the reference procedure. The framework does not assume that this direction corresponds to correctness or optimality; rather, it provides a consistent coordinate system for analyzing how decisions vary with contrast along each axis.

Because all axes are defined over the same set of procedures, comparisons across axes can be performed consistently within the same set of trials.

To connect decisions with structural contrasts, we define an alignment indicator

$$y^{(k)}(\tau) = \mathbf{1}\big[a(\tau) = p_k(\tau)\big]. \tag{9}$$

This indicator converts each trial into a binary outcome that records whether the reuse judgment aligns with the directional comparison induced by axis $k$.

Importantly, alignment does not represent correctness, optimality, or successful reuse. Rather, it records whether the observed decision follows the direction of variation defined by the structural axis under consideration.

Aggregating this indicator over trials enables us to characterize how consistently decisions align with the axis as a function of contrast magnitude $x_k(\tau)$.

The variable $y^{(k)}(\tau)$ therefore serves as a measurement variable that relates comparative reuse judgments to controlled structural variation along axis $k$.

## 2.3 Output Characterization

For each reference procedure $r_q$ and axis $k$, we define the response function

$$f_{r_q}^{(k)}(x) = \mathbb{P}\big(y^{(k)}(\tau) = 1 \mid x_k(\tau) \in \mathcal{B}(x), r_q\big), \tag{10}$$

where $\mathcal{B}(x)$ denotes a bin centered at $x$, used to discretize the contrast variable. The probability is taken over trials $\tau$ with reference procedure $r_q$ and nonzero contrast (i.e., $x_k(\tau) > 0$).

The function $f_{r_q}^{(k)}(x)$ describes the probability that the decision-maker's reuse judgments align with the directional comparison induced by axis $k$ as the structural contrast between candidates increases.

Rather than summarizing judgments using a single scalar, we treat the full response function as a primary object of measurement, capturing how comparative reuse judgments vary as a function of structural contrast along each axis.

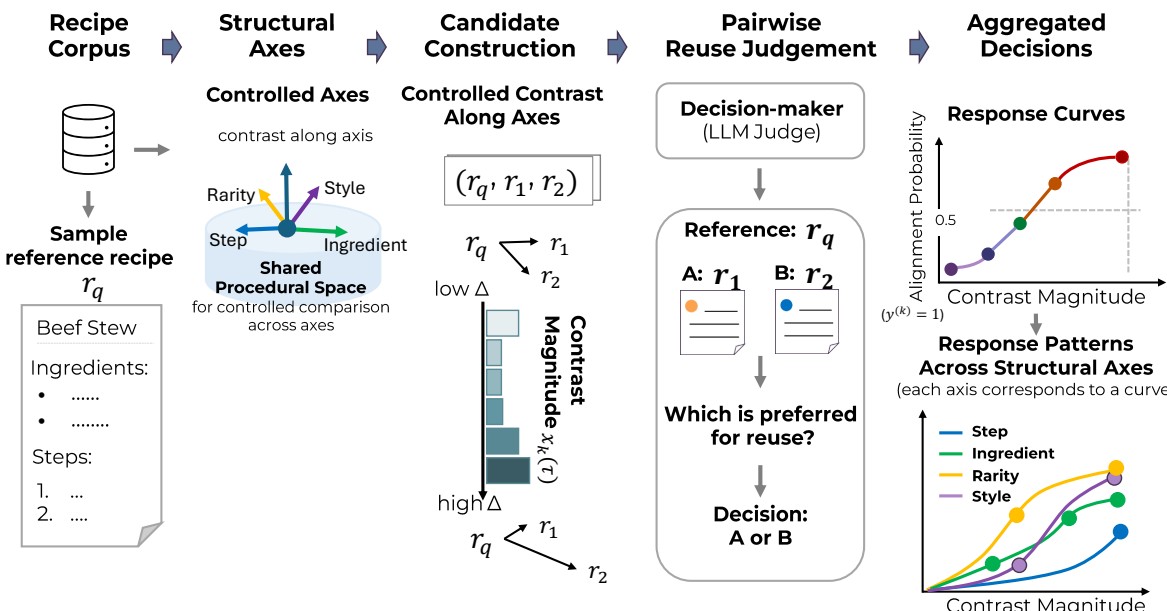

Figure 1: Overview of the measurement pipeline. From a corpus of procedural texts, a reference procedure $r_q$ is sampled and candidate procedures are constructed under controlled contrasts along multiple structural axes defined within a shared procedural space. Pairs of candidate procedures $(r_1, r_2)$ are compared with respect to the reference using a decision-maker. Aggregating these pairwise decisions across bins of contrast magnitude $x_k(\tau)$ yields response curves that characterize how alignment probability varies as a function of contrast magnitude, enabling comparison of response patterns across axes.

The resulting response functions can be compared across reference procedures, structural axes, contextual conditions, and decision-makers.

Such comparisons make it possible to analyze how comparative reuse judgments vary across these factors under controlled variation.

This formulation therefore provides a descriptive characterization of comparative reuse judgments, enabling systematic analysis of how comparative reuse judgments respond to different forms of procedural contrast.

## 3 Experimental Setup

This section describes how the measurement framework introduced in Section 2 is instantiated in our experiments. Following the pipeline illustrated in Figure 1, we describe each stage of the measurement process: starting from the recipe corpus, defining comparison axes, constructing candidate procedures, obtaining pairwise reuse judgments, and aggregating decisions into response curves. Implementation details of dataset construction, candidate sampling, and prompt design are provided in Appendices A and B.

The experiments examine how comparative reuse judgments vary under controlled procedural contrasts along predefined axes, and how the resulting response-curve structures change under contextual interventions.

### 3.1 Procedural Domain

We instantiate the framework in the domain of structured cooking recipes. The recipe corpus is derived from the Food.com dataset, which contains approximately 520,000 user-contributed recipes.

From this dataset we construct a working set of approximately 20,000 recipes through cluster-based sampling to maintain diversity of procedural patterns.

Each recipe corresponds to a procedural artifact $r \in \mathcal{R}$, consisting of a title, an ingredient list, and a sequence of procedural steps.

For each trial, one recipe is selected as the reference procedure $r_q$.

From the sampled corpus, 300 recipes are randomly selected as reference procedures using a fixed dataset split seed. The remaining recipes constitute the candidate pool from which candidate procedures are constructed.

The same reference–candidate split is used across pairing seeds; pairing seeds only affect candidate-pair sampling within this fixed split.

### 3.2 Comparison Axes

We define a set of comparison axes that capture distinct aspects of variation between recipes (e.g., step semantic similarity, ingredient similarity, ingredient rarity, and recipe style). Each axis induces a reference-relative comparison between a reference recipe and candidate recipes, which defines the contrast magnitude used in the analysis, as formalized in Section 2.

These axes specify how contrast magnitude is defined and allow analysis of how comparative reuse judgments vary with respect to contrast magnitude along each axis.

The axes are defined over existing recipes to ensure that the resulting variations correspond to naturally occurring procedural differences.

Implementation details for each axis are provided in Appendix C.

### 3.3 Candidate Construction

Given a reference recipe $r_q$, candidate recipes are compared with the reference along each axis to obtain axis-induced comparison values. Pairwise comparison trials are then constructed by selecting two candidate recipes $r_1$ and $r_2$, forming a trial $\tau = (r_q, r_1, r_2)$.

To enable controlled comparison across axes by holding the set of trials fixed, we adopt a *shared-pairs design* in which the same candidate pairs are reused across structural axes.

For intervention and cross-model analyses, the same candidate pairs are additionally reused across conditions. This reduces variation arising from differences in the underlying procedural comparisons and enables more direct comparison of response-curve structures across contexts and decision-makers.

In the main experiments, candidate pairs are generated using the step semantic axis. The choice of sampling axis serves only to define the distribution of candidate pairs and is held fixed across axes; it is not treated as privileged in the subsequent analysis.

Alternative pair construction strategies were also considered and are reported as diagnostic analyses in Appendix D. These induce different trial distributions and can lead to different response patterns, reflecting the dependence of response curves on the underlying set of candidate pairs.

For this reason, the shared-pairs design is adopted in the main analysis to enable controlled comparison across axes.

In the shared-pairs design, approximately 40 candidate pairs are sampled for each reference procedure. Across the 300 reference procedures used in the main experiments, this yields approximately 12,000 pairwise comparison trials per run.

To assess robustness to candidate-pair sampling, the shared-pairs construction is repeated using three independent pairing seeds. Unless otherwise noted, reported results aggregate over these independent runs.

### 3.4 Reuse Judgment via a Controlled Decision-Maker

For each trial $\tau = (r_q, r_1, r_2)$, a decision-maker selects which candidate is preferred for reuse with respect to the reference recipe.

In the main experiments, this decision-maker is implemented using a large language model (LLM), *Llama-3.1-8B-Instruct* (Grattafiori et al., 2024), under fixed prompting conditions.

The decision-maker is provided with the procedural steps of the reference and candidate recipes. In the baseline condition, only step sequences are shown, without additional metadata such as titles or ingredient lists, and this condition is used for comparison with contextual interventions.

The role of the decision-maker is not to provide a ground-truth assessment of reuse quality, but to provide an observable signal for comparative evaluation under controlled conditions. The analysis focuses on how decisions vary with respect to contrast magnitude $x_k(\tau)$, rather than on the absolute correctness of individual decisions.

Decisions are mapped to alignment indicators following the formulation in Section 2, where alignment means that the selected candidate is consistent with the comparison induced by the corresponding axis (i.e., the candidate indicated by the axis-induced comparison is selected).

To reduce positional bias, the order of candidate recipes is randomly swapped across trials with equal probability (0.5), ensuring that an alignment probability of 0.5 corresponds to chance-level agreement with the axis-induced comparison.

For comparison, we additionally replicate the response-curve pipeline using *Qwen2.5-7B-Instruct* (Qwen et al., 2025) under the same candidate construction, aggregation, and intervention settings. The Qwen results are used to examine the extent to which the observed response patterns depend on the choice of decision-maker.

### 3.5 Null Control Construction

To assess whether observed response patterns can arise from the aggregation procedure alone, we construct two label-level null controls.

In the *random-decision null*, the alignment indicator $y^{(k)}(\tau)$ is replaced by an independent Bernoulli(0.5) sample for every trial. This corresponds to a decision-maker whose responses are independent of the procedures being compared.

In the *shuffled-decision null*, the observed alignment indicators are randomly permuted within each run and axis while preserving the marginal distribution of alignment indicators. This removes the relationship between alignment and contrast magnitude while preserving the overall alignment rate.

In both null controls, reference procedures, candidate pairs, structural axes, contrast values, bins, and the aggregation pipeline are kept fixed. Only the alignment outcomes are randomized.

### 3.6 Contextual Interventions

To examine how comparative reuse judgments change under different task contexts, we introduce contextual interventions by modifying the prompts presented to the decision-maker.

These interventions are evaluated relative to the baseline condition described above and alter the information or instructions provided to the decision-maker while keeping the underlying candidate pairs fixed. The same candidate pairs are reused across baseline and intervention conditions.

We consider the following conditions:

- **Goal specification** (including recipe titles to make task goals explicit)

- **Novice reader** (framing the task for a less experienced user)

- **Time-limited context** (introducing constraints on preparation time)

- **Limited equipment** (restricting available tools or resources)

These interventions introduce different constraints under which reuse decisions are made. Differences in response curves across conditions indicate systematic differences in comparative reuse judgments under these contextual modifications.

Implementation details of prompt modifications are provided in Appendix B.

### 3.7 Aggregation and Statistical Estimation

For each trial and axis, the contrast magnitude $x_k(\tau)$ between the two candidate procedures relative to the reference is computed as defined in Section 2. Trials are grouped into six quantile bins based on the realized contrast magnitudes $x_k(\tau)$. These aggregation bins are distinct from the fixed band values used during candidate-pool construction.

Within each bin, the alignment probability is estimated as the empirical mean of $y^{(k)}(\tau)$ over trials in the bin, corresponding to the fraction of decisions that select the candidate consistent with the axis-induced comparison.

The resulting binned estimates constitute the response curves used throughout the empirical analyses.

## 4 Empirical Results

We analyze comparative reuse judgments using response curves constructed from aggregated pairwise decisions, as defined in Section 2. Each response curve characterizes the alignment probability as a function of contrast magnitude for a given structural axis.

Unless otherwise stated, all results are reported under the steps-only baseline condition described in Section 3, in which the decision-maker is provided only with procedural steps. Results obtained under contextual interventions are then compared against this baseline.

The analysis proceeds in five stages. We first establish whether the observed response curves contain structure beyond randomized controls. We then examine how these response patterns vary across structural axes and reference procedures. Next, we investigate how contextual interventions modify response patterns and whether similar structures remain observable across different decision-makers.

Specifically, we address the following questions:

- (R1) Aggregate response patterns: Do aggregate response curves exhibit structure beyond randomized controls, and how do these patterns differ across structural axes?

- (R2) Reference-level variability: How much heterogeneity exists across reference procedures?

- (R3) Contextual sensitivity: How do contextual interventions alter response patterns?

- (R4) Model dependence: To what extent are the observed response patterns dependent on the choice of decision-maker?

### 4.1 R1: Aggregate Response Curves and Null Controls

Figure 2 compares the observed response curves with two randomized controls. In the random-decision null, alignment outcomes are replaced by independent Bernoulli(0.5) samples. In the shuffled-decision null, observed alignment outcomes are randomly permuted within each run and axis while preserving the overall alignment rate.

Across structural axes, the observed curves differ from the two null controls in different ways. In some axes, such as step semantic and ingredient similarity, the shuffled-decision null remains approximately flat around the overall alignment rate, whereas the observed curves vary more systematically across contrast bins. In several axes, departures from the null controls become more visible toward higher contrast bins, although the strength of this effect varies across axes.

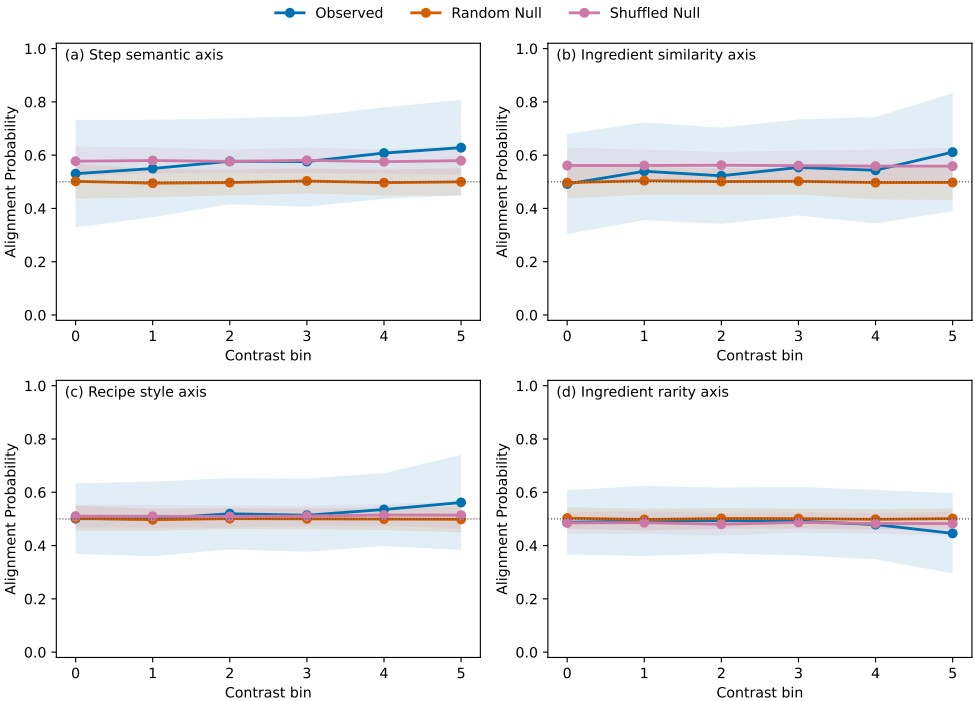

Figure 2: Observed response curves compared with two label-level null controls. For each structural axis, the blue curve shows the observed alignment probability, the orange curve shows the random-decision null, and the pink curve shows the shuffled-decision null. Both null controls preserve the response-curve construction pipeline while randomizing alignment outcomes. The observed curves exhibit patterns that differ from those obtained under the two randomized controls across multiple axes.

These comparisons suggest that the observed response curves are not fully explained by chance-level decisions or by global alignment tendencies alone. Instead, the relationship between alignment probability and contrast magnitude retains axis-dependent structure, with different structural axes exhibiting different response-curve patterns under a common sampling condition.

Because all axes are evaluated over a shared set of candidate pairs, differences between the resulting curves can be compared under a common sampling condition.

These aggregate patterns should be interpreted in the context of the shared-pairs construction used in this study, where candidate pairs are generated using the step semantic axis. Additional axis-specific analyses without the shared-pairs constraint are reported in Appendix D.

Overall, the results suggest that the observed response curves contain structure that is not fully explained by the randomized controls considered here, and that the relationship between alignment probability and contrast magnitude differs across structural axes under a common sampling condition.

Having characterized these aggregate patterns, we next examine how much variability remains at the level of individual reference procedures.

## 4.2 R2: Reference-Level Variability in Response Curves

The aggregate response patterns identified in R1 summarize behavior across many reference procedures. We next examine how much variability exists across individual reference procedures and how this variability relates to the aggregate patterns observed in R1.

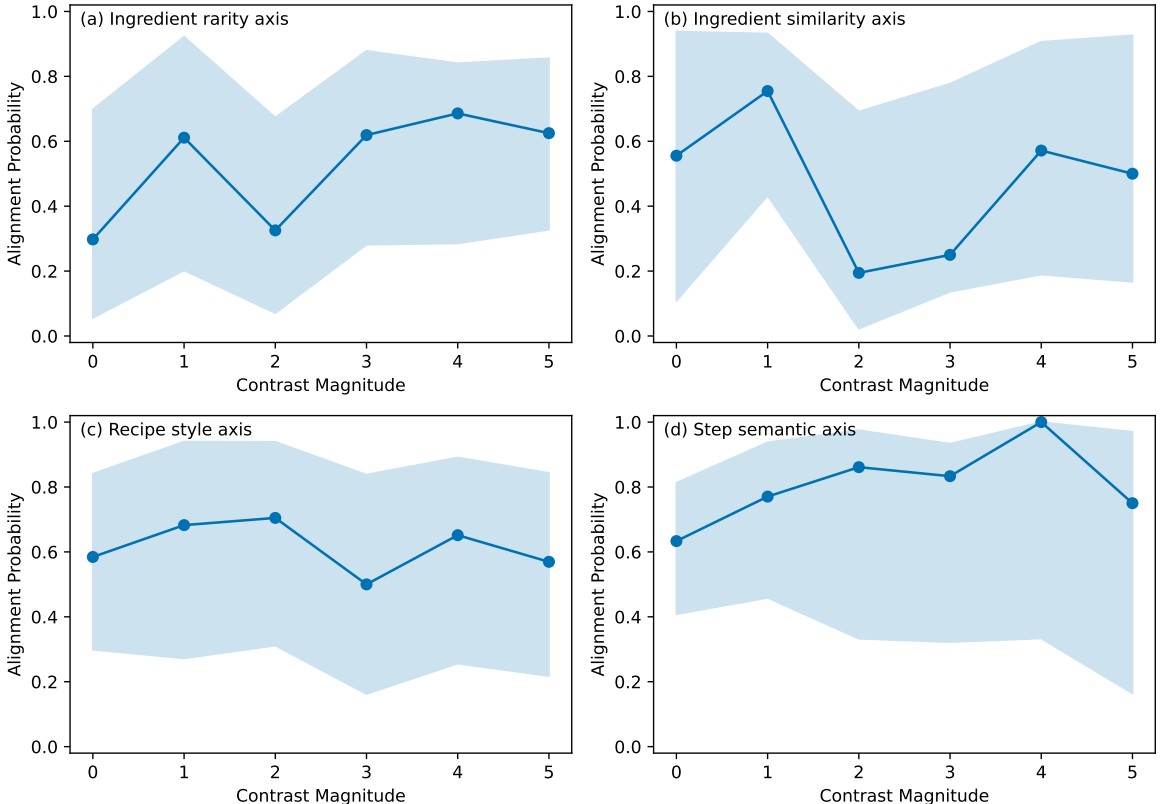

Figure 3: Response curves for an example reference procedure across the four structural axes. Individual response curves often exhibit substantial fluctuations and non-monotonic behavior, illustrating variability at the level of individual reference procedures.

Figure 3 shows response curves for an example reference procedure across the four structural axes. Unlike the aggregate curves, individual response curves often exhibit pronounced fluctuations and non-monotonic behavior. Similar contrast magnitudes may therefore be associated with different alignment probabilities depending on the reference procedure under consideration.

These examples suggest that substantial variability remains at the level of individual reference procedures, even when aggregate response patterns are visible at the population level.

To provide an exploratory visualization of response-curve variability, each response curve is represented as a vector over contrast bins and projected into a two-dimensional space using principal component analysis (Appendix G).

Figure 4 shows the resulting embedding. Response curves occupy a largely overlapping region of the space, without clear separation into a small number of distinct groups. Although weak axis-dependent tendencies are visible, considerable overlap remains across axes.

The projection is broadly consistent with the diversity of response-curve patterns observed across individual reference procedures. However, the visualization is not intended as a quantitative measure of axis-level separation. The observed structure depends on the chosen response-curve representation and dimensionality reduction method, and should not be interpreted as a definitive geometric characterization of reuse judgments.

Taken together, these results indicate that aggregate response curves provide useful population-level summaries, while meaningful variability remains at the level of individual reference procedures. The aggregate

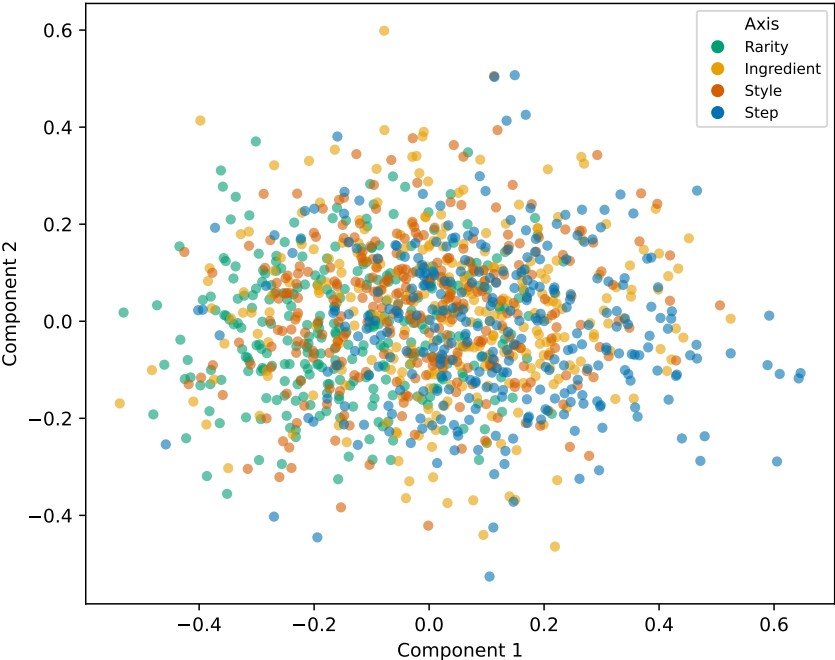

Figure 4: Distribution of response curves in a two-dimensional PCA projection. Each point corresponds to a response curve associated with a particular reference procedure and structural axis.Considerable overlap between axes shows that response curves occupy a largely shared region of the projected space.

Table 1: Displacement between baseline and intervention conditions. For each reference procedure, response curves are constructed under both conditions using the same set of candidate pairs under the step semantic axis. Each response curve is represented as a vector over contrast bins, and displacement is defined as $1-$ cosine similarity between the baseline and intervention curves. Values report mean $\pm$ standard deviation of displacement aggregated over 300 reference procedures.

| Condition | Displacement |
|---|---|
| Goal specification | $0.270 \pm 0.114$ |
| Novice profile | $0.238 \pm 0.109$ |
| Limited equipment | $0.390 \pm 0.150$ |
| Time limited | $0.384 \pm 0.164$ |

axis-level patterns identified in R1 should therefore be interpreted as population-level tendencies rather than behaviors that are uniformly shared across reference procedures.

### 4.3 R3: Intervention-Induced Changes in Response Curves

We next examine whether contextual interventions are associated with measurable changes in response curves relative to the baseline condition.

For each reference procedure, we construct response curves under both the baseline condition and each intervention condition using the same set of candidate pairs under the step semantic axis. Each response curve is represented as a vector whose entries correspond to alignment probabilities over contrast bins.

To quantify changes in response curves, we define displacement as the distance between the baseline and intervention response curves, computed as $1-$ cosine similarity between these vectors. This provides a measure of the degree of difference between baseline and intervention response curves.

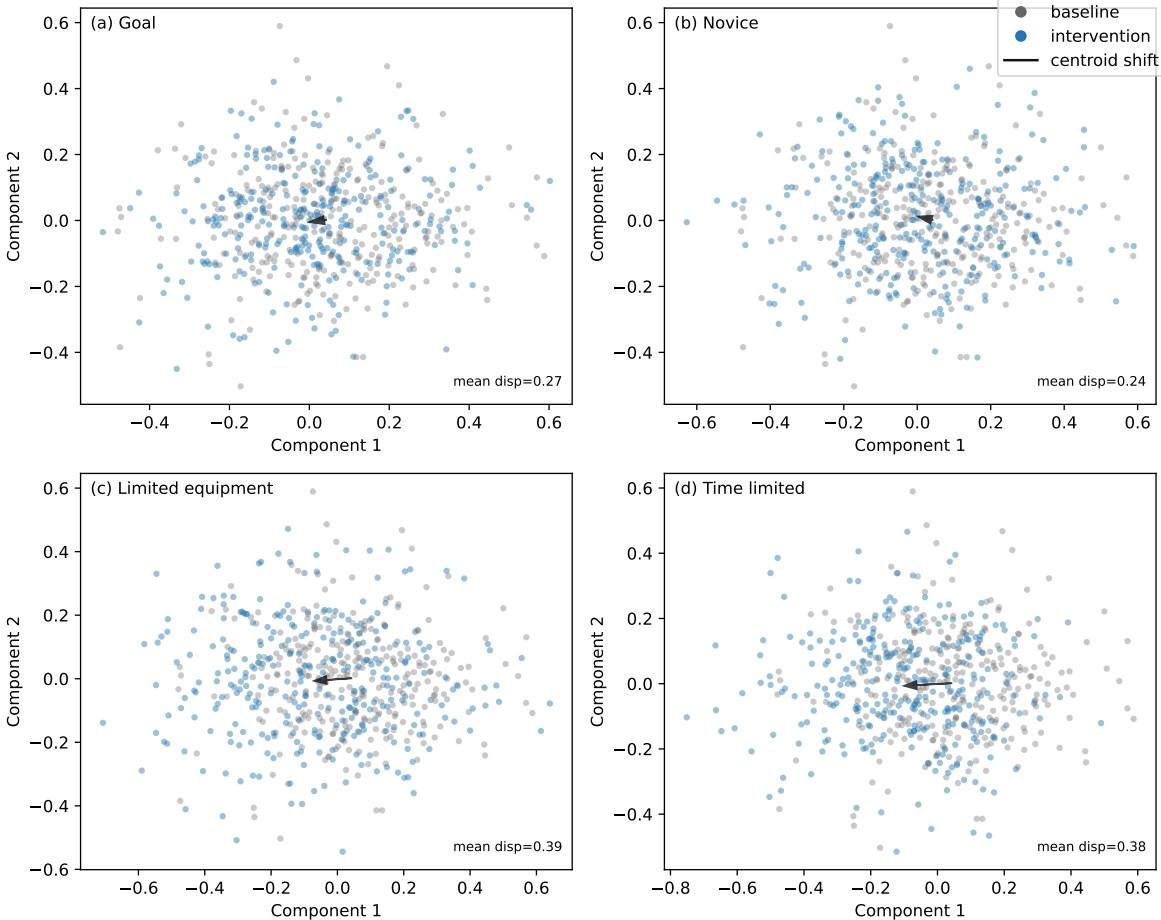

Figure 5: Shifts in response curves under contextual interventions. Each point corresponds to a response curve for a given reference procedure under the step semantic axis, embedded in a low-dimensional space constructed as in Figure 4. Gray points indicate baseline conditions, and blue points indicate intervention conditions. Arrows show the shift in the centroid (mean position) of the distribution from baseline to intervention. Each panel corresponds to a different intervention condition. The visualization provides an exploratory summary of relationships between baseline and intervention response curves. Substantial overlap remains between conditions, although differences in response-curve structure are visible.

For visualization, Figure 5 shows a low-dimensional embedding of response curves under the baseline and intervention conditions. Each point corresponds to a response curve for a given reference procedure, and arrows indicate changes in the mean position of the embedded distribution. The visualization is intended as an exploratory summary of response-curve relationships and is not used for the quantitative measurements reported below.

Across intervention conditions, substantial overlap remains between baseline and intervention response curves, indicating that contextual modifications do not induce complete separation of response patterns. At the same time, visual differences can be observed between the baseline and intervention distributions.

To quantify these changes, Table 1 reports displacement values computed in the original response-curve space rather than in the low-dimensional embedding.

Across all intervention conditions, non-zero displacement is observed between baseline and intervention response curves. The magnitude of displacement varies across intervention types: limited-equipment and time-limited contexts are associated with larger average displacements than goal-specification or novice-profile

contexts. This suggests that the magnitude of intervention-induced changes may differ across intervention types.

Because the candidate pairs remain fixed across conditions, the observed differences cannot be attributed to changes in the underlying set of procedural comparisons. Instead, they are consistent with differences in how the same procedural contrasts are evaluated under different contextual assumptions.

Taken together, these results suggest that contextual interventions are associated with measurable shifts in response-curve structure, with the magnitude of these shifts varying across intervention types. At the same time, substantial overlap with the baseline distribution remains, indicating that contextual information modifies existing response patterns rather than producing completely separate response regimes.

These results suggest that the response-curve framework can be used to characterize context-dependent changes in comparative reuse judgments through shifts in response-curve structure.

Bootstrap confidence intervals for the aggregate response curves are provided in Appendix E. The resulting confidence bands remain broadly consistent with the qualitative trends reported in the main analysis.

### 4.4 R4: Model Dependence of Response Patterns

Finally, we examine the extent to which the observed response patterns depend on the choice of decision-maker.

All analyses are replicated using Qwen2.5-7B-Instruct while keeping the candidate pairs, structural axes, aggregation pipeline, and intervention settings fixed. The goal is not to establish identical reuse judgments across models, but rather to assess whether similar response-curve structures emerge across different model families.

Figure 6 compares the aggregate response curves obtained from Llama-3.1-8B-Instruct and Qwen2.5-7B-Instruct under the baseline condition.

Across all structural axes, both models produce aggregate response curves with qualitatively similar overall patterns. Differences remain visible in both alignment levels and contrast sensitivity, particularly for the step semantic axis, where Llama exhibits a stronger increase in alignment probability across contrast bins.

Taken together, these results suggest that individual reuse judgments and detailed response characteristics remain model-dependent. At the same time, response-curve analysis reveals both shared and model-dependent aspects of comparative reuse judgments when different judge models are evaluated on identical candidate pairs.

These observations suggest that the response-curve framework can be used to characterize differences in comparative reuse judgments across decision-makers under matched procedural and contextual conditions.

Additional comparisons across all intervention conditions are provided in Appendix F.

**Summary**   Response curves exhibit structure beyond randomized controls and show distinct aggregate response patterns across structural axes (R1), while also exhibiting variability across individual reference procedures (R2). Contextual interventions are associated with measurable changes in response patterns (R3), while response-curve analysis enables comparison of comparative reuse judgments across different judge models under matched conditions (R4).

Taken together, these results suggest that response curves provide a common measurement representation for characterizing and comparing comparative reuse judgments across structural, contextual, and model-related dimensions.

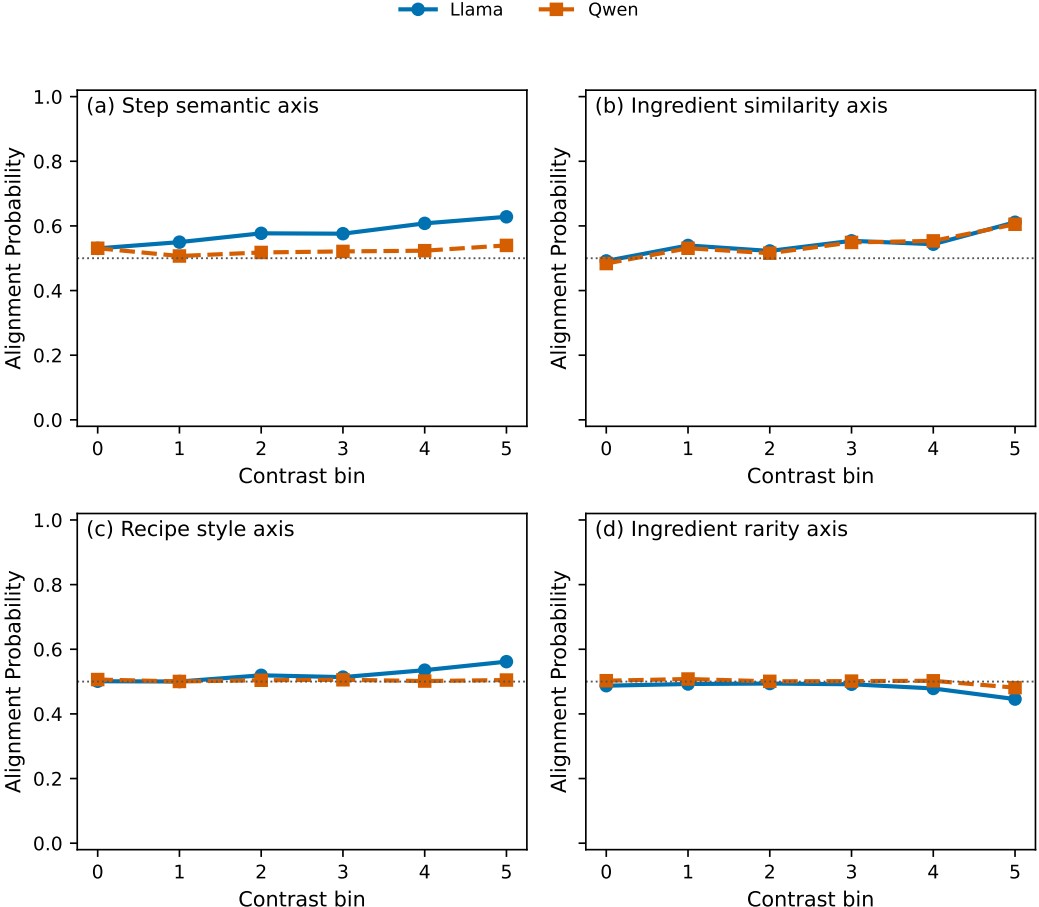

Figure 6: Comparison of aggregate response curves obtained using two different judge models, Llama-3.1-8B-Instruct and Qwen2.5-7B-Instruct, under the baseline condition. For each structural axis, alignment probabilities are averaged across reference procedures within contrast bins using identical candidate pairs and the same aggregation pipeline for both models. The two models exhibit broadly similar aggregate response patterns across all axes, although differences remain visible in both alignment levels and contrast sensitivity, particularly for the step semantic axis.

## 5 Interpreting Response-Curve Structure

The empirical results in Section 4 show that comparative reuse judgments exhibit structured variation across reference procedures, structural axes, contextual conditions, and decision-makers. These patterns are captured by response curves that summarize alignment probability as a function of contrast magnitude.

In this section, we discuss how response curves can be interpreted and what types of insights can be obtained from this representation of comparative reuse judgments.

### 5.1 Reuse Judgments as Response Functions

The comparison with randomized controls (R1) suggests that response curves contain structured variation beyond that expected from the null models considered in this study.

The aggregate response patterns identified in R1 differ across structural axes, while individual reference procedures exhibit considerable variability and often non-monotonic response profiles (R2).

Taken together, these observations suggest that comparative reuse judgments may not be adequately summarized by a single aggregate preference measure. Instead, they are more naturally viewed as response functions that relate structural contrast to alignment probability. Response curves provide a direct representation of this relationship.

## 5.2 Interpretation Gaps as Response-Curve Shifts

The intervention analysis (R3) shows that response curves constructed under different contextual conditions exhibit measurable shifts while retaining substantial overlap with the baseline distribution.

Because these curves are constructed using identical candidate pairs, the observed differences are unlikely to be attributable to changes in the underlying comparisons. Instead, they are consistent with differences in how structural contrasts are evaluated under different contextual assumptions.

Under this perspective, an interpretation gap can be operationalized as a difference in response-curve structure between conditions. Such gaps can be measured through changes in alignment probability across contrast levels, providing a descriptive characterization of how comparative reuse judgments vary under context.

## 5.3 Implications for Analyzing Reuse Judgments

Traditional evaluations of procedural systems often focus on execution outcomes or task success. The present framework instead treats comparative reuse judgments themselves as the object of measurement.

Under this perspective, response curves provide a common representation for analyzing how reuse judgments vary under controlled procedural and contextual variation. Rather than asking whether a procedure can be successfully executed, the framework asks how comparative reuse judgments vary under controlled structural and contextual variation.

The empirical results illustrate three types of structure that become observable through the response-curve representation. First, aggregate response curves reveal reproducible axis-dependent tendencies in comparative reuse judgments, while individual reference procedures often exhibit highly variable and non-monotonic response profiles. This suggests that comparative reuse judgments contain both population-level structure and substantial reference-level heterogeneity.

Second, contextual interventions modify response-curve structure while preserving broad overlap with the baseline distribution. This suggests that contextual information does not simply replace existing judgment patterns, but instead reshapes how structural contrasts are interpreted under different conditions.

Third, although individual reuse judgments remain model-dependent, broadly similar aggregate response structures are recovered across different judge models. This suggests that response-curve analysis can characterize both shared and model-dependent aspects of comparative reuse judgments across different decision-makers.

More broadly, the proposed framework provides a descriptive approach for studying comparative reuse judgments without requiring a predefined notion of optimal reuse judgments.

The empirical patterns reported in this study should therefore be viewed primarily as examples of the types of structure that can be made observable through the framework, rather than as domain-specific conclusions about procedural reuse.

By aggregating large numbers of pairwise decisions into a common response-curve representation, the framework enables systematic comparison of reuse judgments across procedures, contexts, and decision-makers under controlled conditions.

## 6 Relation to Prior Work

Our work relates to research on pairwise comparative evaluation, preference modeling, transferability estimation, and reuse in LLM-based systems. The key distinction of our approach is to treat pairwise judgments

as a measurement process and to analyze how responses vary under controlled structural contrasts, rather than to aggregate comparisons into scalar scores, predictive models, or optimization objectives.

Across these research directions, pairwise comparisons are typically used either to estimate latent scores, learn reward functions, predict transfer outcomes, or improve system performance. In contrast, our objective is descriptive rather than predictive or optimization-oriented. We treat comparative reuse judgments themselves as the object of measurement and use response curves to characterize how such judgments vary under controlled procedural contrasts and contexts.

## 6.1 Pairwise Judgments as Measurement

Pairwise comparisons are widely used to evaluate model outputs and align systems with human preferences (Li et al., 2024; Zheng et al., 2023; Chiang et al., 2024). Classical models such as Thurstone, Bradley–Terry, and Elo aggregate such comparisons into latent scalar scores or rankings (Thurstone, 1927; Bradley & Terry, 1952; Elo, 1978).

In contrast, we do not aim to recover a latent score. Instead, we treat comparative reuse judgments elicited through pairwise decisions as measurements and analyze how they vary as a function of contrast magnitude. Under this perspective, pairwise comparisons are used to estimate a contrast–response relationship, rather than to fit a ranking model. This perspective is related to measurement approaches in psychophysics, where response probabilities are analyzed as a function of controlled stimulus differences (Green & Swets, 1966; Wichmann & Hill, 2001).

## 6.2 From Preference Modeling to Response Characterization

Recent work on preference learning and reward modeling uses human preference data, often collected as rankings or comparisons between outputs, to infer latent reward functions or preference models, for example in reinforcement learning from human feedback (RLHF) settings (Christiano et al., 2017; Ouyang et al., 2022). In these approaches, preference data are typically aggregated into a scalar reward signal used for optimization.

In contrast, our objective is not to infer a reward function or preference model from comparisons. Instead, we use comparative reuse judgments as observations for characterizing how response behavior varies under controlled procedural contrasts. This enables direct characterization of response structure, whereas scalar reward modeling primarily focuses on learning a latent reward signal for optimization.

## 6.3 Task Similarity and Transferability

Prior work on transferability estimation studies how knowledge learned for one task generalizes to another, using representations and metrics such as Task2Vec, LEEP, and LogME (Achille et al., 2019; Nguyen et al., 2020; You et al., 2021).

These approaches aim to predict downstream transfer performance. In contrast, we do not predict transfer outcomes. Instead, we measure how comparative reuse judgments vary under controlled structural contrasts. The axes in our framework are not predictive features but serve as measurement coordinates for probing comparative reuse judgments.

More generally, these approaches rely on task representations and similarity measures to characterize relationships between tasks. Our framework similarly depends on axis-specific procedural representations, but uses them as measurement coordinates for analyzing comparative reuse judgments rather than for predicting transfer performance.

## 6.4 Reuse in LLM-Based Systems

Recent work on LLM-based agents explores mechanisms for storing and reusing past experience, intermediate reasoning, or retrieved context (Wang et al., 2023; Shinn et al., 2023; Lewis et al., 2020).

These approaches focus on enabling or improving reuse within LLM-based systems. In contrast, our objective is not to design a reuse mechanism but to provide a framework for measuring and analyzing comparative reuse judgments under controlled variation.

Whereas reuse systems focus on storing, retrieving, or adapting past experience, our framework focuses on how alternative procedures are comparatively evaluated for reuse with respect to a reference procedure and a given context.

The two directions are therefore complementary: reuse mechanisms determine how prior experience is stored, retrieved, or adapted, while our framework characterizes how alternative procedures are comparatively evaluated for reuse.

# 7 Limitations and Scope

Our framework is designed to make reuse judgments observable under controlled variation. However, several limitations remain.

## 7.1 LLM-Based Evaluation and Scope

We use LLMs as decision-makers to instantiate the framework. We do not assume that the resulting judgments correspond to human decisions; rather, they provide an observable signal that can be systematically analyzed within the proposed framework. Extending the framework to human evaluators remains an important direction.

The framework does not define an optimal reuse strategy. Instead, it measures how alignment probabilities vary across controlled procedural contrasts. Observed shifts should therefore be interpreted as changes in the measured response patterns of comparative reuse judgments, rather than improvements in reuse quality.

Consequently, the empirical results should be interpreted as characterizing the comparative reuse judgments of the evaluated decision-makers rather than general properties of human reuse judgments.

## 7.2 Axis Design and Dependencies

The structural axes are heuristic and not intended as validated predictors of transfer success. Their role is to provide controlled coordinates for measuring how alignment probabilities vary with contrast magnitude.

In the present study, these axes are manually specified based on domain knowledge. Alternative axis definitions or automatically derived representations may reveal different response structures.

Candidate procedures are constructed using a shared-pairs design in which sampling is performed along a specific axis (step semantic similarity). As a result, the distribution of candidate pairs is not uniform across axes, and observed response patterns along other axes are conditioned on this sampling procedure.

This introduces a dependency between axis definition and data distribution. In particular, axis-dependent differences in response curves should be interpreted as measurements under the given sampling condition, rather than as intrinsic properties of the axes.

The structural axes are defined over a shared procedural representation and may exhibit correlations with one another. For example, procedures that are similar in step structure may also share ingredients or stylistic properties. As a result, variations along one axis may partially co-vary with others, making it difficult to attribute response changes to a single axis in isolation. Disentangling such dependencies would require jointly controlled or orthogonalized axis constructions.

More generally, the framework is descriptive rather than causal. Response curves characterize associations between controlled procedural contrasts and comparative reuse judgments, but do not by themselves establish causal mechanisms underlying the observed response patterns.

### 7.3 Estimation and Discretization Effects

Response curves are constructed by grouping trials into bins based on contrast magnitude and estimating alignment probabilities within each bin. As a result, the shape of the response curves depends on the choice of binning scheme, including bin boundaries and the number of bins.

While our analysis uses a fixed binning strategy for consistency, alternative discretizations may yield different curve shapes, particularly in regions with sparse data. Exploring binning-free or adaptive estimation methods remains an important direction for future work.

### 7.4 Domain and Coverage Limitations

We focus on cooking recipes as a structured procedural domain. While the framework may generalize to other domains, such generalization is not evaluated here.

We also consider a limited set of contextual interventions. Future work may explore a broader space of constraints and how they reshape response curves.

## 8 Conclusion

We introduced a measurement framework for studying comparative reuse judgments through response curves defined over controlled procedural contrasts.

By constructing controlled pairwise comparisons, the framework treats comparative reuse judgments as observable outcomes and characterizes how alignment probabilities vary as structural contrast increases along different axes.

Empirically, response curves exhibit structure beyond randomized controls, differ across structural axes, show substantial heterogeneity across reference procedures, and undergo measurable changes under contextual interventions. Although individual judgments remain model-dependent, broadly similar aggregate response structures are recovered across different judge models.

These findings suggest that comparative reuse judgments are more naturally characterized as response functions over structural contrast rather than as single similarity scores or global preference measures.

More broadly, the empirical patterns reported in this study illustrate several types of structure that become observable through the proposed framework, including population-level response patterns, reference-level heterogeneity, context-dependent shifts, and model-dependent differences.

By aggregating pairwise decisions into a common response-curve representation, the framework enables systematic comparison of comparative reuse judgments across procedures, contexts, and decision-makers under controlled conditions.

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

## A  Experimental Setup

### A.1  Dataset and Reference Sampling

The experiments use a recipe corpus derived from the Food.com dataset Food.com, which contains approximately 520,000 user-contributed recipes.

From this dataset, we construct a working set of approximately 20,000 recipes using cluster-based sampling to ensure diversity of procedural patterns.

From the sampled corpus, 300 recipes are randomly selected as reference procedures ($q\_size = 300$) for the main experiments.

### A.2  Candidate Pair Construction

Candidate pairs are constructed using a shared-pairs design so that the same set of trials is used across all structural axes.

For each reference procedure $r_q$, candidate recipes are drawn from the sampled corpus. In the main experiments, candidates are ranked according to their step-semantic contrast magnitude with respect to $r_q$. This axis is used only for constructing candidate pairs and does not affect the definition of other structural axes used in the analysis.

Candidates are grouped into bins according to their contrast magnitude values along the step-semantic axis, and candidate pairs are randomly sampled within each bin to form trials $\tau = (r_q, r_1, r_2)$.

In the shared-pairs experiments, approximately 40 candidate pairs are sampled per reference procedure ($pairs\_per\_reference = 40$). These same pairs are reused across all structural axes. For intervention and cross-model analyses, the same candidate pairs are additionally reused across conditions within each pairing seed.

This design ensures that all axes are evaluated on the same underlying set of trials. As a result, differences in response curves across axes can be examined under a common sampling condition, reducing variation arising from differences in candidate sampling.

For diagnostic experiments (Appendix D), candidate pairs may instead be constructed separately for each axis by ranking candidates according to the corresponding axis. These axis-specific constructions are used to examine response patterns within individual axes and to assess whether qualitative trends observed under the shared-pairs design remain visible under alternative sampling procedures. They are not used for the main analysis.

### A.3 Contrast Binning

Fixed band values are used during candidate-pool construction. For the step semantic, ingredient similarity, and recipe style axes, the candidate-pool bands are

$$\{0, 2, 4, 6, 8, 10\}.$$

For ingredient rarity, the candidate-pool bands are

$$\{-10, -6, -2, 0, 2, 6, 10\}.$$

These fixed bands are used to construct candidate pools and sample candidate pairs. They are not used directly as the response-curve bins.

For response-curve estimation, we compute the realized contrast magnitude for each candidate pair and partition these values into six quantile-based bins using the empirical distribution within each axis. The x-axis values shown in the main response-curve figures therefore correspond to contrast-bin indices $0, \ldots, 5$, rather than to the raw fixed band values.

### A.4 LLM Judge Configuration

The primary experiments use the Llama-3.1-8B-Instruct model as the judge.

Inference settings are:

- temperature: 0.0
- top_p: 1.0
- max tokens: 64
- seed: 1

To reduce positional bias, the order of candidate procedures is randomly swapped with probability 0.5.

## B Prompt Templates

Reuse judgments are elicited through pairwise comparison prompts presented to the LLM judge.

Each prompt consists of three components:

- an instruction describing the reuse task,

- a reference procedure (the "current recipe"),

- two candidate procedures labeled *A* and *B*.

In the experimental framework described in the main text, the choice between candidates corresponds to the decision variable $a(\tau) \in \{1, 2\}$.

## B.1 Baseline Prompt

The baseline instruction used in the experiments is shown below.

> You will compare two candidate procedural texts (A and B).
>
> Below is the CURRENT RECIPE you are cooking (your own dish). You want to quickly borrow and adapt part of one candidate (a step, technique, or idea) to improve your CURRENT RECIPE.
>
> Which candidate feels more reusable for improving your CURRENT RECIPE?
>
> Output a one-line JSON object with keys "choice" and "confidence". "choice" must be either "A" or "B". "confidence" must be a number between 0.0 and 1.0.

Recipes are rendered primarily as procedural step sequences. Depending on the experimental condition, additional contextual information (e.g., recipe-title information) may also be included.

The final prompt presented to the judge has the following structure:

> [Instruction]
>
> — CURRENT RECIPE (your own dish you are trying to improve) — [Reference procedure text]
>
> — CANDIDATE A — [Candidate procedure A]
>
> — CANDIDATE B — [Candidate procedure B]

## B.2 Prompt Modifications for Intervention Conditions

Contextual interventions described in Section 3 are implemented by modifying either the instruction or the information included in the rendered procedure text.

In particular, goal specification is introduced by including the recipe title in the rendered procedure text, while other interventions such as novice, time-limited, and limited-equipment conditions are implemented by augmenting the instruction with additional context describing the assumed scenario.

**Example instruction modifications** Interventions implemented through instruction changes are realized by prepending fixed templates that specify the assumed context. For example:

> **Novice:** You are a novice home cook. You prefer straightforward, low-risk adaptations.
>
> **Time-limited:** You have only 20 minutes to finish your current recipe. You must prioritize ideas that reduce total cooking time and avoid techniques requiring long preparation.

Other intervention conditions follow a similar pattern, using fixed instruction templates to express constraints or assumed capabilities.

All prompts preserve the same overall structure as the baseline, and the exact wording of each intervention template is kept fixed across all trials.

## C  Structural Axis Definitions

All structural axes instantiate the comparison function $\Delta_k$ defined in Section 2, and are computed over procedures drawn from the same corpus, ensuring consistency with the shared procedural space.

For all axes, $\Delta_k(r_i, r_j)$ is defined as a scalar contrast measure. Depending on the axis, it may represent either a symmetric distance or a signed difference, and is used consistently within each axis to construct contrast magnitude.

**Step semantic axis** Procedural steps are embedded using a SentenceTransformers model (`all-MiniLM-L6-v2`) (Reimers & Gurevych, 2019). Let $E(r)$ denote the set of step embeddings for procedure $r$. The comparison function is defined as:

$$\Delta_{\text{step}}(r_i, r_j) = \frac{1}{|E(r_i)|} \sum_{e \in E(r_i)} \min_{e' \in E(r_j)} \big(1 - \cos(e, e')\big).$$

This defines a directed semantic distance from $r_i$ to $r_j$ based on step-level nearest-neighbor matching. This computes, for each step in $r_i$, the distance to its nearest semantically similar step in $r_j$, and averages these distances across all steps.

**Ingredient similarity axis** Ingredient lists are converted into normalized tokens with stopword removal, and represented using a presence-based TF–IDF scheme (i.e., token occurrence without counting multiplicity). Let $v_r$ denote the resulting TF–IDF vector for recipe $r$. The comparison function is defined as:

$$\Delta_{\text{ingredient}}(r_i, r_j) = 1 - \cos(v_{r_i}, v_{r_j}).$$

This defines a symmetric distance between ingredient sets.

**Ingredient rarity axis** Let $\tilde{I}(r)$ denote the set of deduplicated normalized ingredient items in recipe $r$. Ingredient rarity is defined using information content:

$$IC(i) = -\log p(i).$$

The rarity score of a recipe is defined as the mean information content over its ingredients:

$$R(r) = \frac{1}{|\tilde{I}(r)|} \sum_{i \in \tilde{I}(r)} IC(i).$$

The comparison function is defined as:

$$\Delta_{\text{rarity}}(r_q, c) = R(c) - R(q).$$

This defines a signed contrast reflecting relative ingredient rarity with respect to the reference procedure.

**Recipe style axis** Each recipe is represented by a feature vector capturing coarse structural properties of procedural text, including counts and statistics over steps and ingredients, length-based features, and indicators such as numeric and temporal expressions. Let $s_r$ denote the resulting feature vector for recipe $r$. The comparison function is defined as:

$$\Delta_{\text{style}}(r_i, r_j) = 1 - \cos(s_{r_i}, s_{r_j}).$$

This defines a symmetric distance over stylistic properties of procedural descriptions.

## D   Per-Axis Response Curves

### D.1   Per-Axis Diagnostic Experiments

To complement the shared-pairs design, we conduct axis-specific experiments in which candidate pairs are constructed separately for each axis.

In these experiments, candidates are ranked according to the corresponding axis, and approximately 50 candidate pairs are sampled per reference procedure ($pairs\_per\_reference = 50$, $q\_size = 300$).

These experiments allow examination of response patterns within each axis independently, providing a complementary view of how reuse judgments vary with contrast magnitude.

Because candidate pairs differ across axes, comparisons across axes are not controlled for differences in the underlying trial set. For this reason, all main analyses involving cross-axis comparisons are based on the shared-pairs design.

### D.2   Per-Axis Response Curves

To complement the shared-pairs analysis reported in R1, we visualize response curves obtained under the axis-specific sampling procedure described above.

Figure 7 shows the resulting response curves. Each gray curve corresponds to a reference procedure, and the black curve indicates the mean.

Compared to the shared-pairs setting, the magnitude and shape of response curves vary due to differences in the underlying trial distributions. Nevertheless, several response patterns observed under the shared-pairs design remain visible in these axis-specific experiments, although their strength and shape may differ across axes.

## E   Bootstrap Confidence Intervals

To characterize uncertainty in the aggregate response curves, we additionally compute 95% bootstrap confidence intervals by resampling reference procedures after averaging across pairing seeds. These intervals complement the variability bands shown in the main text, which reflect variation across reference procedures.

For each condition, axis, and reference procedure, response curves are first averaged across pairing seeds. Reference procedures are then resampled with replacement, preserving the original number of references, and the aggregate response curve is recomputed. This process is repeated for 1000 bootstrap samples. Pairwise trials are not resampled.

Figure 8 shows the resulting confidence intervals for the baseline and intervention conditions. Overall qualitative trends are broadly consistent with those reported in the main analysis, supporting the robustness of the aggregate response patterns to variation in the sampled reference procedures.

## F   Additional Model Comparison

To complement the baseline comparison reported in R4, Figure 9 shows aggregate response curves obtained using Llama-3.1-8B-Instruct and Qwen2.5-7B-Instruct across all intervention conditions.

Across intervention settings, the two models continue to exhibit broadly similar response patterns, although differences in alignment levels and contrast sensitivity remain visible for some axes and conditions.

These results provide additional evidence that the two judge models often exhibit similar qualitative response patterns across intervention conditions, while differing in alignment levels and contrast sensitivity for some axes and contexts.

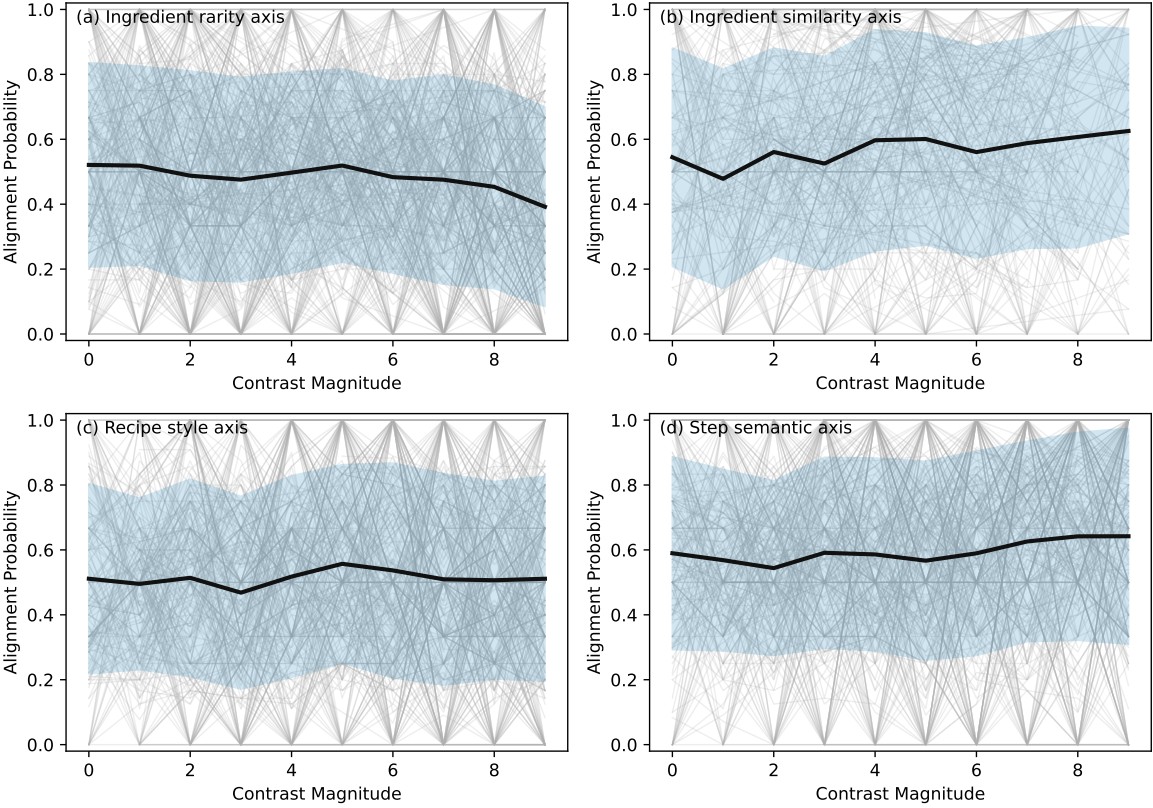

Figure 7: Per-axis response curves under axis-specific sampling. Each gray curve corresponds to a reference procedure, and the black curve indicates the mean across reference procedures.

## G  Response-Curve Embedding

Response curves are represented as vectors whose entries correspond to alignment probabilities over contrast-magnitude bins.

For visualization, response-curve vectors are first computed separately for each pairing seed under the baseline condition. For each reference–axis pair, the vectors are then averaged across pairing seeds to obtain a single representation used for PCA.

The first two principal components are used to obtain a two-dimensional projection for visualization. The embedding is used only for visualization purposes and does not affect any quantitative measurements reported in the main text.

## H  Seed Stability

As an additional robustness check, we examine the sensitivity of response curves to the random pairing seed used in the shared-pairs design.

For each reference procedure and axis, response curves are constructed under three independent pairing seeds. Each response curve is represented as a vector over contrast bins, and similarity is quantified using cosine similarity between curves obtained from different seeds.

Figure 10 (left) shows example response curves for a single reference procedure. Individual bins can exhibit noticeable variability across seeds, indicating that response curves are locally sensitive to differences in candidate pairings.

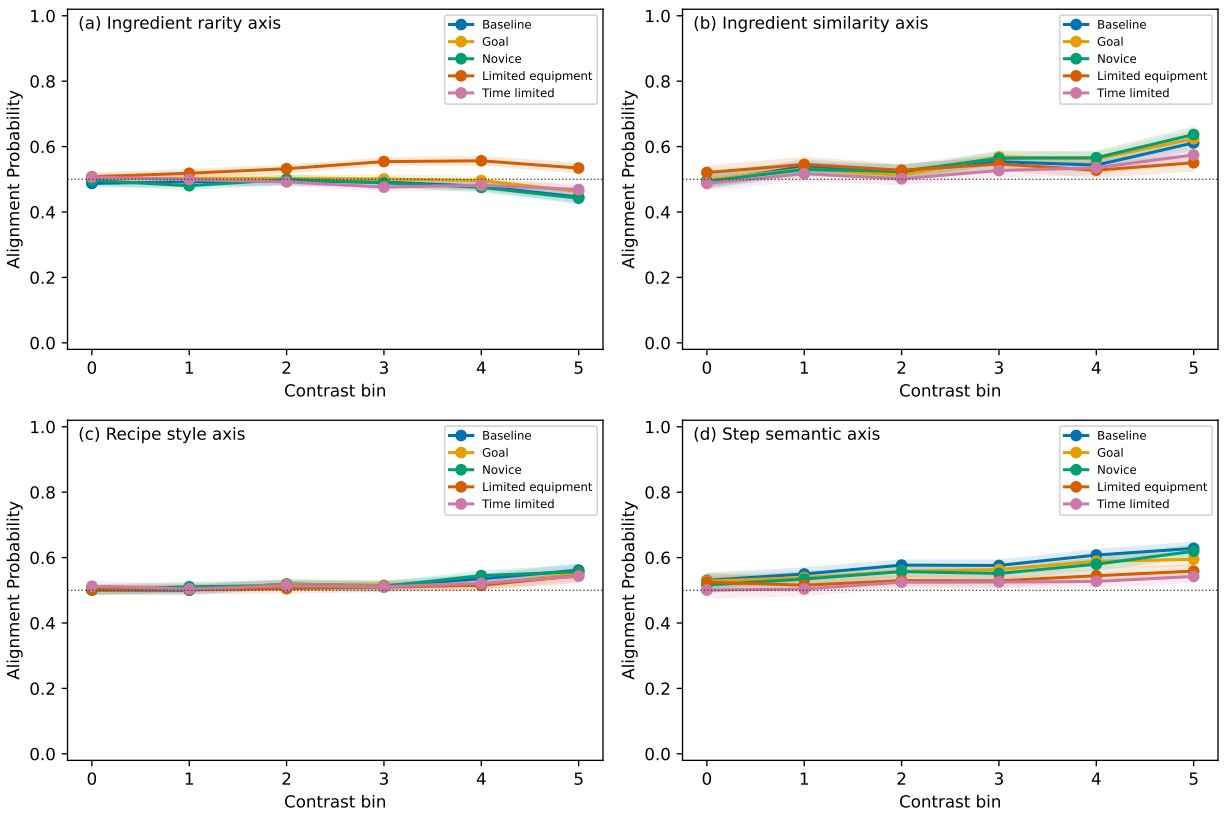

Figure 8: Aggregate response curves with 95intervals obtained by resampling reference procedures after averaging across pairing seeds. Rows correspond to structural axes and colors correspond to experimental conditions. The overall qualitative trends are broadly consistent with those reported in the main analysis.

Table 2: Seed-wise similarity of response curves. Response curves are represented as vectors over contrast bins, and cosine similarity is computed between curves obtained from different pairing seeds for the same reference procedure and axis. Values report mean ± standard deviation aggregated across reference–axis combinations. For seed-stability analysis, the number of references may be smaller than 300 when curve similarity cannot be computed for all pairing-seed pairs for a given reference procedure and axis.

| Axis | Cosine similarity | #References |
|------|-------------------|-------------|
| Step semantic | $0.881 \pm 0.095$ | 300 |
| Ingredient similarity | $0.828 \pm 0.133$ | 290 |
| Ingredient rarity | $0.871 \pm 0.085$ | 300 |
| Recipe style | $0.868 \pm 0.090$ | 300 |

To quantify this variability, Figure 10 (right) shows the distribution of pairwise cosine similarities between curves obtained from different pairing seeds, aggregated across reference procedures and axes. The distribution is concentrated toward higher similarity values, with a median cosine similarity of 0.889.

Table 2 summarizes these similarities as mean ± standard deviation for each axis across reference procedures.

Although local fluctuations are observed, response curves remain broadly similar across independent pairing seeds, with mean cosine similarities ranging from 0.828 to 0.881 across axes. These results suggest that the overall shape of the response curves is not strongly dependent on a particular random pairing configuration.

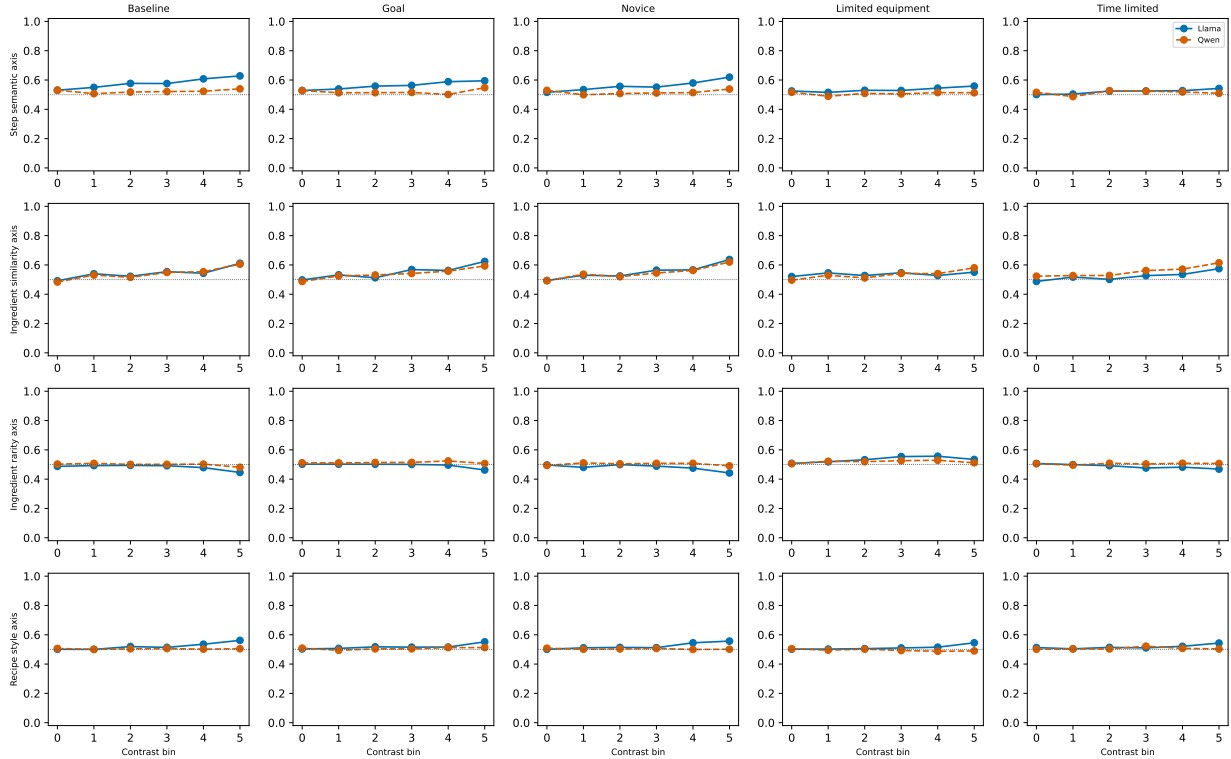

Figure 9: Aggregate response curves obtained using Llama-3.1-8B-Instruct and Qwen2.5-7B-Instruct across all intervention conditions. Rows correspond to structural axes and columns correspond to experimental conditions. Although local differences remain visible, broadly similar response patterns are observed across the two model families under a wide range of intervention settings.

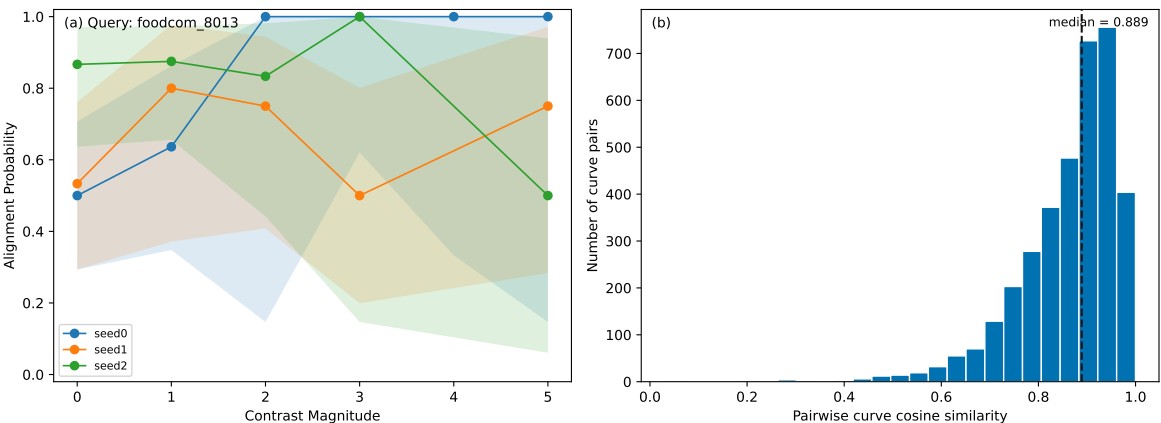

Figure 10: Sensitivity of response curves to pairing seeds. For each reference procedure and axis, response curves are constructed under three independent pairing seeds using the shared-pairs design. Left: example response curves for a single reference procedure. Right: distribution of pairwise cosine similarities between response curves obtained from different seeds, aggregated across all reference–axis combinations.

