# OpenReview forum: "Measuring Procedural Reuse Judgments with Response Curves"
_TMLR — Rejected by TMLR_

### Review · Reviewer_hdPH · 2026-05-10

**Summary Of Contributions:**

The authors propose to analyze the utility of procedural knowledge by formulating procedural alternatives and comparing their relative fitness for adaptation and reuse. Here, the authors introduce the concept of 'reuse judgments', the pairwise comparison of alternative formulations of cooking recipes by judging which of the alternatives yields a better fit for a particular task under consideration. Under the proposed reuse judgments, alternative recipes are analyzed under 4 controlled aspects (e.g., recipe style, semantic similarity ...). Alternatives are created and judged with the use of an LLM. Finally, the authors empirically evaluate their procedure under aspects of stability, axis dependence, response curve distribution and contextual interventions over a set of 300 sample recipes.



**Strengths**

1. **Setup.** The authors present an interesting and intuitive idea for their paper. The approach is well motivated and described. The initial problem setup and method of measurement is well laid out. Figure 1 aids understanding of the overall proposed procedure and experimental setup.

2. **Presentation.** The paper is well organized and structured. The analysis of results follows a clear path and are presented with detailed discussions and interpretations of the obtained results. The presented research question structure the findings and guide clearly through the individual evaluations.



**Weaknesses**

1. **Significance of Results.** In addition to the use of a single LLM the authors, conduct their analysis over only a 300 data point subset from an available set of 20,000 recipes. Considering the various plots in the main paper and appendix, the data seems to feature high variance/noise, with relatively small effect sizes. While the authors acknowledge the high variance and 'local fluctuations' across individual samples and bins, their analysis does not incorporate any any significance tests or p-values to substantiate their findings. Similarly, the authors describe binning in Appendix A.3 with bins in steps of two and exclude bins with insufficient samples. Given the amount of potentially available data, the analysis could easily be strengthened and bins be more fine-grained if more samples were used.
2. **Evaluation and Data.** Another major weakness of the paper is the use of a single Llama-3.1-8B model for evaluation. Given the rapid progress in LLMs this model might be considered rather outdated and weak in terms of performance, hindering significance of results. Similarly, while the authors provide the prompts used in their evaluation, no sample data or rephrased recipes are provided. Considering the strong reliance on LLMs for the evaluation and rewriting, the paper does not provide enough insight to properly check the well working of the proposed pipeline.
3. **Writing.** The paper could be improved in terms of writing. The formatting of the paper into sequences of many small paragraphs (even having the the abstract consist of 4 paragraphs) is rather uncommon and makes the paper tiresome to read. Most of the paper consists of prose text, making it wordy and in part imprecise. Apart from the 10 formulas in section 2, that rather regard evaluation metrics than experimental details, the paper contains few technical details or formula in the main body. In particular the interpretation of evaluations could be strengthened by providing further significance results.

**Audience:**

No

**Audience Explanation:**

I generally like the idea and detailed discussion of results in the paper. My main concerns however regard the soundness of results and the drawn conclusions. As it is not clear whether particular results hold beyond the limited evaluation of the paper, I do not believe that the paper in its current state would be of interest to the TMLR audience.

**Claims And Evidence:**

No

**Claims Explanation:**

As outlined in the weaknesses above, the paper lacks in rigor with respect to the data analysis, lack variations on LLM and features a relatively small sample size in the absence of any significance tests.

**Use of LLM.** As outlined before. Experiments are conducted with a rather weak and, in terms of LLM development, rather 'old' Llama-3.1-8B-Instruct model. While the task might not require strong mathematical or general reasoning skills, the use of a single model limits judgment and weakening significance as no further, stronger models or other model families are compared that might give rise to other distributions or less noisy judgements. Given that the authors are trying to demonstrate reuse judgments in the general case beyond the particular tested model, it is unclear how results would vary.

**Significance and Interpretation of results.** Tables and figures in the paper seem to admit different conclusions with regard to the significance of results. While tables 1 and 2 seem to present high cosine similarities and displacements under low variance, the low-dimensional projections such as shown in figures  5 and 6 seem to show almost no visible signal or separation of data. I'd question if any of these plots admit for a meaningful interpretation of results or whether the visible shifts might simply be due to noise. While embeddings might drastically skew distributions, I'd question interpretations like "distributions of response curves [...] exhibit consistent shifts in their centroid positions".

Similarly, figure 7 seems to feature strong alignment of individual lines to to either 0.0 or 1.0 alignment, disclosing strong noise and variance in the data. While the claimed similarities results in the main paper could possibly be reflected for subplots a and b, subplots c and d seem to show trends in the main figure but not in the appendix. Again, considering the high-variance in contrast to the minor trends, I'd question the actual presence of trends.

**Requested Changes:**

The requested changes mainly regard the weaknesses as outlined above:

1. **Use of Data.** While even the full dataset might not permit for detecting a conclusive trend, the current evaluation with only 300 samples out of a potentially available 20.000 samples features high variances and leaves too much room for uncertainty. The concerns, as mentioned above, could be tackled by simply using more of the available data, beyond the currently used 300 samples.
2. **Significance of Results.** The paper need to be more concise in quantifying the significance in relation to noise. As already outlined, the authors should provide significance results to substantiate their results and exclude the interpretation of noise artifacts.
3. **Use of LLM.** The authors demonstrate that their results hold up under varying LLM.
4. **Code and Data.** The authors should provide accompanying code and include samples to enable checking the quality of the pipeline and the rephrased data.
5. **Minor.** The authors might want to consider restructuring their paper into fewer consistent paragraphs.

---

> ### Author Response · Authors · 2026-05-25
> **Additional analyses for robustness, uncertainty, and model dependence**
>
> Thank you for the detailed and constructive review. We especially appreciate the concerns regarding evaluation scale, uncertainty, model dependence, and the interpretation of potentially weak or noisy response structures.
> We agree that several aspects of the original manuscript required stronger statistical characterization and more careful interpretation. To address these concerns, we conducted additional analyses aimed at better distinguishing systematic response behavior from noise artifacts.
>
>
> First, we clarified the effective evaluation scale. The “300 recipes” correspond to reference procedures rather than individual datapoints. For each reference procedure, approximately 40 candidate-pair trials are constructed, resulting in roughly 12,000 pairwise reuse judgments in the main experimental setting.
>
>
> Second, following the reviewer’s concern regarding possible noise artifacts, we implemented additional null-baseline analyses using both randomized and shuffled decision controls while preserving the original candidate pairs and aggregation pipeline. Across intervention conditions, observed response-curve displacements remained consistently larger on average than those obtained under either null control. Because shuffled controls preserve marginal label statistics, these analyses help distinguish observed structure from purely noise-driven behavior or label imbalance effects.
>
>
> Third, we computed bootstrap confidence intervals by resampling reference procedures rather than individual trials in order to better characterize uncertainty at the level of procedural variation. These analyses indicate substantial heterogeneity across queries, but also show systematic deviations from null behavior, particularly along the step-semantic axis. Together with the null-baseline controls, these analyses help assess whether the observed response-curve shifts remain distinguishable from noise-driven behavior under resampling and control conditions.
>
>
> We additionally expanded the cross-model analysis beyond the original appendix comparison with Qwen. Our intention is not to claim that different models produce identical reuse judgments. Rather, the goal is to examine whether qualitatively related response structures remain observable across different judge models while still allowing substantial model-dependent variability. We found that similar response-curve organization and intervention-sensitive shifts can also be observed under Qwen, although with non-identical response behavior.
>
>
> We also agree that some interpretations in the original manuscript may have been overly strong relative to the observed effect sizes. In revision, we will recalibrate the framing to more clearly emphasize the exploratory and measurement-oriented nature of the framework, the substantial overlap and heterogeneity in response curves, and the interpretation of the observed patterns as weak but measurable response structures rather than strong universal reuse laws.
>
>
> Finally, we appreciate the comments regarding presentation and reproducibility. In revision, we will improve the organization of the manuscript, consolidate fragmented paragraphs, expand methodological details, and include additional examples and implementation clarifications to facilitate inspection of the rewriting and judgment pipeline.

---

### Review · Reviewer_oKb3 · 2026-05-13

**Summary Of Contributions:**

*Summary*

This paper proposes a procedural reuse judgments, i.e., given a reference procedure and two candidate procedures, a LLM chooses which candidate is more reusable. The authors define structural axes such as step similarity, ingredient similarity, ingredient rarity, and recipe style, then aggregate pairwise LLM judgments into response curves showing alignment probability as a function of contrast magnitude. Empirical results are over food recipe task.


---

*Concerns*

1. I am concerned that if the results in this paper is statistical significant or not. I mean the error band is too wide, and it also makes me very hard to interpret the results (e.g., Fig 6).
2. I mean for judges re-use, LLM-as-a-judge should be the most important one task to study here (e.g., AlpacaEval, Arena-Hard). Why study food recipe? The motivation for choosing cooking recipes is underdeveloped. If the paper aims to study procedural reuse generally, it should either justify why recipes are a representative procedural domain or evaluate on an additional domains.
3. I think the judge often can't see the information used by some axes.
4. I mean statistically, the empirical evaluation should be stronger. At least include some hypothesis testing. The current empirical evaluation reads somehow confusing and seems handy-crafted.

**Additional Comments:**

I tried my best to understand this paper, but given the unfamiliarity to author's aim, I found the paper’s high-level goal and evaluation target difficult to assess. The work appears to propose a new measurement framework rather than improve a prior baseline, but the practical purpose of the framework and its relation to existing LLM evaluation or procedural-reuse tasks are not clearly established. I ask the AE to take this into account when weighing my review.

**Audience:**

Yes

**Audience Explanation:**

If it can be extended to more general response re-use measurement, then there should be some audiences interested in.

**Claims And Evidence:**

No

**Claims Explanation:**

For my understanding, I still don't understand how good or bad LLM responds to controlled prompts when comparing to the actual human procedural reuse judgments. If the results mainly characterize prompt-conditioned LLM preference behavior, then I would suggest to make it clear as the scope of the paper.

**Requested Changes:**

I think it should take some effort to revise the paper or clarify the concerns. Please check the concerns in the Summary of Contribution section and resolve them.

---

> ### Author Response · Authors · 2026-05-25
> **Clarification of scope, procedural reuse framing, and statistical characterization**
>
> Thank you for the thoughtful review and for highlighting concerns regarding the evaluation target, statistical interpretation, and the relationship between the proposed framework and existing LLM-as-a-judge literature.
>
>
> We agree that the original manuscript did not sufficiently clarify the intended scope of the work. In particular, the paper is not intended as a human-alignment benchmark or as an attempt to recover a ground-truth notion of optimal procedural reuse. Rather, the goal is to operationalize and measure comparative reuse judgments under controlled procedural contrasts, and to characterize how these judgments vary across structural and contextual conditions.
>
>
> More generally, our use of LLMs differs from standard LLM-as-a-judge settings in which the model primarily acts as an evaluator approximating a target notion of correctness or human preference. In our framework, the model instead serves as a decision-making observer whose comparative responses under controlled procedural contrasts constitute the primary object of analysis. Accordingly, the framework does not assume that the model explicitly recovers the engineered axes as discrete symbolic variables. Rather, the axes define controlled contrast manipulations used to probe how comparative applicability judgments respond under different procedural variations.
>
>
> We will also clarify more explicitly why the framework focuses on reuse rather than generic preference. In our formulation, reuse judgments are reference- and context-conditioned: the judged applicability of a candidate depends on the current procedure being adapted and the constraints under which adaptation is considered. The objective is therefore not to identify the globally preferred recipe, but to analyze comparative applicability judgments regarding which candidate appears more reusable as an adaptation source under a particular context.
>
>
> Relatedly, we agree that the motivation for focusing on recipes should be strengthened. Our intention was not to claim that cooking recipes uniquely represent all procedural domains, but rather that recipes provide a structured procedural setting with naturally occurring goals, resources, constraints, and adaptation scenarios. This makes them suitable for constructing controlled procedural contrasts and studying context-conditioned reuse judgments in a concrete and interpretable domain.
>
>
> Regarding the reviewer’s concerns about noise and statistical interpretation, we conducted several additional analyses to better characterize uncertainty and distinguish observed structure from null behavior.
> First, we clarified the effective evaluation scale. The “300 recipes” correspond to reference procedures rather than individual datapoints. For each reference procedure, approximately 40 candidate-pair trials are constructed, resulting in roughly 12,000 pairwise reuse judgments in the main setting.
>
>
> Second, we implemented additional null-baseline analyses using both randomized and shuffled decision controls while preserving the original candidate pairs and aggregation pipeline. Across intervention conditions, observed response-curve displacements remained consistently larger on average than those obtained under either null control. Importantly, shuffled controls preserve marginal label statistics, helping distinguish observed structure from purely noise-driven behavior.
>
>
> Third, we computed bootstrap confidence intervals by resampling reference procedures rather than individual trials in order to better characterize uncertainty at the level of procedural variation. These analyses indicate substantial heterogeneity across queries, while still showing systematic deviations from null behavior, particularly along the step-semantic axis.
>
>
> We additionally expanded the cross-model analysis beyond the original appendix comparison with Qwen. We do not expect identical response structures across models; rather, the goal is to examine whether qualitatively related response patterns remain observable across different judge models while allowing model-dependent variability.
>
>
> Finally, we agree that several interpretations in the original manuscript would benefit from more cautious framing. In revision, we will recalibrate the presentation to better emphasize the exploratory and descriptive nature of the framework and to avoid overstating the strength or universality of the observed response structures.

---

> > ### Comment · Reviewer_oKb3 · 2026-05-27
> >
> > Did you update the manuscript? TMLR encourages update new draft during rebuttal.

---

> > > ### Author Response · Authors · 2026-05-27
> > >
> > > Thank you for the clarification. We are currently incorporating the revisions into an updated manuscript and plan to upload the revised draft within the next few days.

---

### Review · Reviewer_8fqi · 2026-05-20

**Summary Of Contributions:**

The authors present an analysis framework for analyzing reuse judgments through controlled pairwise comparisons.

**Audience:**

Yes

**Audience Explanation:**

The method itself has interesting characteristics. I can definitely see future work building on the techniques presented here, if the manuscript is strengthened.

**Broader Impact Concerns:**

I have no broader impact concerns that would require further discussion.

**Claims And Evidence:**

No

**Claims Explanation:**

In my opinion, the biggest weakness of the manuscript is the lack of a precise problem statement. The reader is left without a crisp understanding of the types of insights that one could obtain from this framework; and it is difficult to asses the efficacy of the proposed analysis without a concrete set of properties that the measurements should satisfy.

List of claims that may need further evidence:

1. The manuscript claims to present a framework for analysis of reuse judgements, but no formal guarantees or descriptions of the analysis are given. This is in my opinion the biggest weakness of the manuscript.
2. The manuscript claims to present an analysis framework designed to study reuse judgements. However, no aspects of the framework itself are tied to reuse specifically. That is, the judgements could be described simply as a stating a preference, and reuse judgements are simply one possible application of the framework. Clarification may be needed needed on why is there a strong emphasis on reuse and what guarantees does the framework provide specific to reuse.
3. The manuscript claims the analysis presents "implications for systems that generate, adapt, or select procedures under varying constraints", but it is unclear what such implications are.

**Requested Changes:**

1. Provide a precise statement of the goal of the analysis and the measurements done by the framework.
2. Add discussion (limitations and comparison to related work) on the implications of featurization.
	-  No related work that relies on featurization to study preferences or comparisons is cited.
	- Featurization introduces a very substantial assumption: that a domain-specific featurizer is available; in general this is non-trivial to engineer. This is in contrast to prior work that learns a latent score (inferred from data). This increases the engineering necessary to apply the proposed framework. On these grounds I suggest that comparison to prior work can be expanded.
3.  Clarify the source of the reference procedure and why it's part of the framework.
	- The stated goal of the framework is to study "reuse judgments, defined as comparative decisions about which of two candidate procedures is preferred for reuse with respect to a reference," but it's never stated why it's necessary to study this with respect to a reference, and why reuse specifically.
4. Include verifiable claims about the effectiveness of the analysis (i.e., does it work? what can we learn from the analysis?). It is difficult from the presented empirical results to understand whether the framework is effective. This is related to a lack of a formal problem statement. E.g., what should we make of the presented empirical results? What should we expect of the measurement values under different data distributions? Are there guidelines on how to interpret the results if we apply the framework to other settings?
5. (Minor) A single LLM is used for the experiments. Could we use the framework to detect systematic differences between different models?

---

> ### Author Response · Authors · 2026-05-25
> **Clarification of reuse-specific measurement framing and reference-conditioned evaluation**
>
> Thank you for the detailed and thoughtful review. We especially appreciate the concerns regarding the conceptual framing of the framework, the role of reuse-specific evaluation, and the interpretation of the resulting measurements.
>
>
> We agree that the original manuscript did not sufficiently clarify the intended scope of the framework. In revision, we will more explicitly position the work as a measurement-oriented framework for characterizing comparative reuse judgments under controlled procedural contrasts, rather than as a predictive model, latent utility estimator, or execution-based evaluation framework.
>
>
> A central point we will clarify is why the framework focuses specifically on reuse judgments rather than generic pairwise preference. In our formulation, reuse is inherently reference- and context-conditioned: the judged applicability of a candidate procedure depends on the current procedure being adapted, as well as the situational context under which adaptation is considered. The goal is therefore not to determine which candidate is globally “better,” but rather which candidate appears more reusable as an adaptation source relative to a current procedure and contextual constraints.
>
>
> Relatedly, we will clarify the role of the reference procedure in the framework. The reference is not simply additional conditioning information, but defines the adaptation target relative to which comparative applicability is evaluated. This formulation is intended to operationalize one observable component of procedural adaptation: comparative judgments regarding what should be borrowed or adapted under a given context.
>
>
> We also agree that the manuscript did not sufficiently distinguish the proposed framework from latent-score or preference-modeling approaches. In revision, we will expand the discussion of related work and more clearly contrast our approach with methods that aggregate pairwise comparisons into scalar utilities or rankings. Our intention is instead to characterize response behavior as a function of controlled contrast magnitude, in a manner more closely related to comparative-judgment and psychophysical measurement paradigms.
>
>
> We also agree that the interpretation and intended use of the resulting measurements were insufficiently clarified in the original manuscript. Our intention is not to treat the resulting measurements as correctness metrics or universal reuse laws, but rather as descriptive characterizations of how comparative applicability judgments vary under controlled procedural contrasts and contextual conditions.
>
> More specifically, our goal is to expose how applicability criteria shift across procedural contrasts, contextual conditions, and decision-makers, rather than collapsing these judgments into a single scalar preference score. We believe this characterization may provide a useful diagnostic perspective for analyzing and inspecting procedural adaptation behavior in generation or recommendation settings. For example, the framework can expose which procedural dimensions remain relatively stable under contextual interventions and which forms of procedural similarity exhibit stronger model-dependent variability.
>
> We will also clarify that the framework is intended as a descriptive measurement protocol rather than a framework with formal optimality or identifiability guarantees.
>
>
> Regarding the interpretation of the empirical results, we agree that several claims would benefit from more careful qualification. In revision, we will recalibrate the framing to better emphasize the exploratory and descriptive nature of the framework, the substantial heterogeneity in response behavior, and the interpretation of the observed structures as weak but measurable response patterns rather than strong universal laws of procedural reuse.
>
>
> We also appreciate the reviewer’s comments regarding axis construction and featurization. In revision, we will expand the limitations and related-work discussion regarding the assumptions introduced by hand-designed structural axes, their dependence on procedural representations, and their relationship to learned latent representations and transferability-oriented approaches.
>
>
> Finally, we agree that the cross-model aspect was insufficiently emphasized in the original manuscript. While the appendix already included an additional judge-model analysis using Qwen, we have expanded this analysis to more explicitly examine similarities and differences in response structures across model families while allowing model-dependent variability.

---

### Decision · Action_Editor_Uy8L · 2026-06-29

**Recommendation:** Reject

**Audience:**

Yes

**Audience Explanation:**

The topic itself is relevant.

**Claims And Evidence:**

No

**Claims Explanation:**

A consensus among the reviewers has been reached to reject this paper. The paper is confusing and claims are imprecise (e.g., no precise problem statement; no verifiable claims about the efficacy of the proposed analysis are stated; conclusions seem overly broad for the evidence presented; structure is unclear). Experimental settings are problematic (only 300 recipes are used despite access to about 20,000 recipes) and results do not look statistically sound (hard to interpret; the initial source of the large observed variance could still not be attributed to the method or the used data conclusively).